# Dual Effects of 3-*epi*-betulin from *Daphniphyllum glaucescens* in Suppressing SARS-CoV-2-Induced Inflammation and Inhibiting Virus Entry

**DOI:** 10.3390/ijms242317040

**Published:** 2023-12-01

**Authors:** Yung-Ju Yeh, Tai-Ling Chao, Yu-Jen Chang, Sui-Yuan Chang, Chih-Hao Lu, Chih-Hua Chao, Wen-Chi Su, Ching-Ping Tseng, Michael M.C. Lai, Ju-Chien Cheng

**Affiliations:** 1Department of Medical Laboratory Science and Biotechnology, China Medical University, Taichung 404328, Taiwan; 2Research Center for Emerging Viruses, China Medical University Hospital, Taichung 404328, Taiwan; 3Department of Clinical Laboratory Sciences and Medical Biotechnology, National Taiwan University, Taipei 100229, Taiwan; 4The Ph.D. Program of Biotechnology and Biomedical Industry, China Medical University, Taichung 404328, Taiwan; 5Department of Laboratory Medicine, National Taiwan University Hospital and National Taiwan University College Medicine, Taipei 100229, Taiwan; 6Institute of Bioinformatics and Systems Biology, National Yang Ming Chiao Tung University, Hsinchu 300093, Taiwan; 7Department of Biological Science and Technology, National Yang Ming Chiao Tung University, Hsinchu 300093, Taiwan; 8School of Pharmacy, China Medical University, Taichung 4060404, Taiwan; 9Department of Medical Research, China Medical University Hospital, Taichung 404328, Taiwan; 10Department of Medical Biotechnology and Laboratory Science, Chang Gung University, Taoyuan 33302, Taiwan; 11Graduate Institute of Biomedical Sciences, China Medical University, Taichung 404328, Taiwan

**Keywords:** 3-*epi*-betulin, SARS-CoV-2, inflammation, virus entry inhibitor

## Abstract

The continuous emergence of SARS-CoV-2 variants has led to a protracted global COVID-19 pandemic with significant impacts on public health and global economy. While there are currently available SARS-CoV-2 vaccines and therapeutics, most of the FDA-approved antiviral agents directly target viral proteins. However, inflammation is the initial immune pathogenesis induced by SARS-CoV-2 infection, there is still a need to find additional agents that can control the virus in the early stages of infection to alleviate disease progression for the next pandemic. Here, we find that both the spike protein and its receptor CD147 are crucial for inducing inflammation by SARS-CoV-2 in THP-1 monocytic cells. Moreover, we find that 3-*epi*-betulin, isolated from *Daphniphyllum glaucescens*, reduces the level of proinflammatory cytokines induced by SARS-CoV-2, consequently resulting in a decreased viral RNA accumulation and plaque formation. In addition, 3-*epi*-betulin displays a broad-spectrum inhibition of entry of SARS-CoV-2 pseudoviruses, including Alpha (B.1.1.7), Eplison (B.1.429), Gamma (P1), Delta (B.1.617.2) and Omicron (BA.1). Moreover, 3-*epi*-betulin potently inhibits SARS-CoV-2 infection with an EC_50_ of <20 μM in Calu-3 lung epithelial cells. Bioinformatic analysis reveals the chemical interaction between the 3-*epi*-betulin and the spike protein, along with the critical amino acid residues in the spike protein that contribute to the inhibitory activity of 3-*epi*-betulin against virus entry. Taken together, our results suggest that 3-*epi*-betulin exhibits dual effect: it reduces SARS-CoV-2-induced inflammation and inhibits virus entry, positioning it as a potential antiviral agent against SARS-CoV-2.

## 1. Introduction

SARS-CoV-2 is the causative agent of the COVID-19 infectious disease, which has spread rapidly worldwide since its identification in late 2019. The virus caused a global pandemic that has had a significant impact on public health and the global economy. Multiple variants of SARS-CoV-2 have been discovered and some of them are more infectious than the original strain, while others may not show a significant increase in transmission. SARS-CoV-2 is a positive-sense, single-stranded RNA virus. The structural proteins of SARS-CoV-2 include membrane glycoprotein (M), envelope protein (E), nucleocapsid protein (N), and the spike protein (S). Among these, the spike protein is the key structural protein for inducing inflammation [1]. Upon infection, the receptor-binding domain (RBD) of the spike protein binds to angiotensin converting enzyme 2 (ACE2) on the surface of target cells, facilitating the subsequent entry of the virus into the cells [2]. In addition to ACE2, other potential receptors have been proposed, such as glucose-regulated protein 78 (GRP78) and CD147 [3,4].

SARS-CoV-2 is highly contagious, leading to either asymptomatic infection or a range of symptoms, including cough, fever, fatigue, muscle aches and headaches, shortness of breath, even pneumonia and acute respiratory distress syndrome (ARDS) [5]. Inflammation is a key feature of the immune response to the virus and plays a significant role in the symptoms and severity of COVID-19 [6,7]. SARS-CoV-2-induced inflammation causing the release of cytokines, such as Interleukin-1 beta (IL-1 beta), Interleukin-6 (IL-6), Tumor necrosis factor-alpha (TNF-alpha), and Interferon-gamma (IFN-gamma), which contribute to the disease severity and death [8]. Hyper-inflammation is considered one of the major drivers of severe COVID-19 and long COVID [9]. Hence, molecules that possess dual antiviral and anti-inflammatory properties are in high demand.

Traditional medicinal plants and their natural product derivatives have been used in preventing and treating a wide range of infectious diseases [10,11,12]. Consequently, some of these natural compounds have been developed as promising candidates for combating SARS-CoV-2 [13,14]. In this study, we discovered that compound 3-*epi*-betulin, isolated from *Daphniphyllum glaucescens*, exhibits a dual anti-SARS-CoV-2 effect. It effectively reduces virus-induced inflammation and acts as a broad-spectrum inhibitor for virus entry in a cell culture model.

## 2. Results

### 2.1. The SARS-CoV-2 Pseudovirus Activates an Inflammatory Response Independent of Virus Entry

To understand whether the structural proteins of SARS-CoV-2 can stimulate cell inflammation, two types of pseudotyped virus particles mimicking SARS-CoV-2 were used. These particles contained either the spike, envelope, and membrane proteins (SEMpv) or the spike protein only (Spv). THP-1 cells were stimulated with these particles, and the inflammatory response was evaluated by detecting the central mediator of inflammation, phospho-NFκB p65, downstream pro-inflammatory genes, and released cytokines. The data showed that both SEMpv and Spv activated NFκB p65 phosphorylation 4.1-fold and 3.9-fold, respectively (Figure 1A). In addition, both SEMpv and Spv induced mRNA levels of pro-inflammatory genes IL1B (2.9-fold vs. 3.1-fold), IL6 (2.01-fold vs. 2.34-fold), and TNF (2.37-fold vs. 1.86-fold), respectively. Similar pseudoviral particles were used to stimulate the inflammatory response, and the cellular viral RNA was evaluated to determine viral infectivity (Figure 1B). Finally, a similar amount of IL-1β cytokine was detected in THP-1 cells stimulated with either SEMpv or Spv (Figure 1C).

To further clarify whether viral entry is important for the activation of inflammation, UV-inactivated pseudotyped viruses were used in the same experiments. Interestingly, the phosphorylation of p65 was still activated by UV-inactivated SEMpv as well as SEMpv (3.2-fold vs. 3.8-fold, respectively) (Figure 2A). Subsequently, the pro-inflammatory genes IL1B (2.82-fold vs. 2.93-fold), IL6 (2.18-fold vs. 2.26-fold), and TNF (2.11-fold vs. 2.06-fold) were also increased by SEMpv or UV-inactivated SEMpv stimulation. However, the intracellular viral RNA in UV-inactivated SEMpv infection was reduced by 66% compared to SEMpv (Figure 2B). Additionally, IL-1β cytokine release was also promoted by UV-inactivated SEMpv stimulation (Figure 2C). Similar data were displayed by UV-inactivated Spv. These data above implied that the inflammation is not related to the virus entering cells. We therefore wonder if the binding of spike to receptor is crucial for stimulating inflammation. Angiotensin-converting enzyme 2 (ACE2) is the most important receptor for SARS-CoV-2 entry [15]. However, CD147 was reported to mediate the virus entering the cells by endocytosis [4]. Hence, the expression levels of ACE2 and CD147 in human lung epithelial Calu-3 and macrophage THP-1 cells were determined by immunofluorescence assay. The results demonstrated that ACE2 was highly expressed in Calu-3 cells but not THP-1 cells. However, CD147 was expressed in both cells, especially in THP-1 cells (Figure 3A). Next, the CD147 was knocked down to estimate the effect of CD147 on spike-stimulated inflammation. The expression of knockdown-CD147 was evaluated by flow cytometry. Compared to the negative control (shGFP), the mean fluorescence intensity (MFI) of knockdown-CD147 (shCD147C, shCD147D) on THP-1 cells was decreased by 45.5% and 46.6%, respectively (Figure 3B). In addition, the spike-stimulated inflammation was reduced significantly in CD147 knockdown cells. The phosphor status of NFκB was decreased by 50% in spike-pseudotyped virus-stimulated CD147 knockdown-THP1 cells (Figure 3C, left panel). In addition, the levels of IL-1B mRNA and secreted IL-1β were decreased by 61% and 50.3%, respectively in spike-pseudotyped virus-stimulated CD147 knockdown-THP1 cells (Figure 3C, middle and right panels). These results implied that spike–CD147 interaction is crucial for SARS-CoV-2-induced inflammation in THP-1 cells. Taken together, we established a system to induce inflammation by SARS-CoV-2 pseudovirus.

### 2.2. Compound ***5*** with Anti-Inflammation Activity Induced by SARS-CoV-2 Pseudoviruses Suppresses Viral Replication

Dinorditerpenes from *Flueggea virosa* are known to exhibit suppressive activity toward the phosphorylation levels of NF-kB P65 in PMA-differentiated THP-1 cells stimulated by SARS-CoV-2 pseudovirus [16]. Additionally, triterpenes are well-known for their role in conferring anti-inflammatory properties to herbal remedies [17]. Thus, compounds of these two types were selected from the two plant sources for the SARS-CoV-2 pseudovirus-induced inflammatory system (Table 1). The differentiated THP-1 cells were infected with SEMpv, followed by treatment with the indicated compound, listed in Table 1, for 16 h. Among these compounds, the IL1B RNA level was significantly decreased, by 47.3%, by compound **5** (cpd. 5, 3-*epi*-betulin) (Figure 4A). To assess the anti-inflammatory activity that may suppress the replication of SARS-CoV-2, we utilized SARS-CoV-2 replicon cells to test the antiviral effects on viral replication. Here we utilized a noninfectious self-replicating SARS-CoV-2 replicon, derived from pBAC-SARS-CoV-2. The pBAC-SARS-CoV-2 contains the 5′ UTR, ORF1a/1b, and 3′UTR of the SARS-CoV-2 genome. Additionally, to facilitate the detection of replication, the firefly luciferase reporter gene was incorporated into pBAC-SARS-CoV-2. To detect the replication of the replicon, HEK293T cells were co-transfected with pBAC-SARS-CoV-2 and pCAG2-NP-HA. At 5 h post-transfection, the transfected cells were reseeded onto a 96-well plate. At 24 h post-transfection, the cells were treated with various concentrations of compound **5**. The luciferase activity, representing the replication activity, was measured at 48 h post-transfection. Compound **5** demonstrated significant dose-dependent inhibition of replication in SARS-CoV-2 replicon-transfected cells. The EC50 was 11.7 ± 0.4 μM (Figure 4B). To confirm the anti-inflammation activity of compound **5**, calu-3 cells were infected with SARS-CoV-2 (hCoV-19/Taiwan/NTU49/2020) at MOI of 0.001 for 1 h. After infection, the cells were treated with either DMSO or 10 μM compound **5** for 24 h. The cellular RNA was collected and the expression of pro-inflammatory genes, IL6 and TNF, was decreased by 66.6% and 46.0%, respectively, according to RT-qPCR (Figure 4C). Consequently, the plus-stranded viral RNA exhibited a reduction of 73.8%, whereas the minus-stranded viral RNA decreased by 51.7% (Figure 4C, right panel). In addition, the culture supernatant viral load decreased by 61.4%, and plaque formation showed a decline of 51.5% (Figure 4D). These data suggest that compound **5** attenuated inflammation induced by SARS-CoV-2 and subsequently reduced viral replication.

To rule out the possibility that compound **5** may inhibit inflammation through the inhibition of viral entry, a time-of-addition study was designed, as shown in Figure 5A. The two upper strategies were presented for inhibiting the distinct step of viral entry, including pre-treatment to target viral receptors and pre-inoculation to disrupt viral binding. In the pre-treatment procedure, compound **5** was initially added to the cells for 1 h, followed by pseudotyped virus infection. In the pre-inoculation procedure, compound **5** was pre-mixed with pseudotyped virus for 1 h before adding the mixture to the cells. The pro-inflammatory genes were detected by RT-qPCR. The expression of pro-inflammatory genes remained unchanged regardless of whether compound **5** was pretreated with differentiated THP-1 cells or mixed with pseudoviruses before infection (Figure 5B,C). In contrast, when compound **5** was administered to differentiated THP-1 cells after pseudovirus infection, there was a 69.5% decrease in IL1B expression and an 82.9% decrease in IL6 expression (Figure 5D). These data indicated that compound **5** inhibited SARS-CoV-2-induced inflammation at the stage of post-infection but not related to viral entry.

### 2.3. Compound ***5*** Docked the RBD Domain of the SARS-CoV-2 Spike Protein and Effectively Inhibited Virus Entry

Since compound **5** belongs to the class of triterpenes, we are curious whether compounds with similar structures possess antiviral activities. Accidentally, we found that the compound betulinic acid, which shares a similar chemical structure with compound **5**, has been reported to inhibit SARS-CoV-2 virus spike binding to ACE2 through the RBD domain [18]. Using the molecular docking method via iGEMDOCK, compound **5** and betulinic acid were observed to anchor with the wild-type SARS-CoV-2 spike proteins. These two compounds are docked into similar positions of the spike protein (Figure 5A) and interact with six residues, which are Y495, G496, Q498, T500, N501, and Y505 (Table 2). Compared to betulinic acid, compound **5** has an extra hydrogen bond interaction with residue Y453 and lower binding energy. We therefore selected compounds with a chemical structure similar to compound **5** from our chemical library, as listed in Table 1, for the SARS-CoV-2 pseudovirus entry assay. To achieve this, SARS-CoV-2 pseudoviruses were pre-treated with either DMSO or the specified compounds at 37 °C for 1 h. Following this, BHK-ACE2 cells were incubated with the compound–virus mixture for an additional 48 h, and luciferase activity was assessed. Consistent with the docking results, compound **5** exhibited a better inhibitory effect on virus entry compared to betulinic acid. Moreover, compound **5** displayed the highest inhibitory activity for virus entry among all the compounds tested and with no cytotoxicity (Figure 6B,C). To further understand the inhibition effect of compound **5** on different variants of SARS-CoV-2, we employed pseudoviruses carrying spike proteins of different variants in the pseudovirus entry assay. The EC50 of each variant of the SARS-CoV-2 pseudovirus entry activity was determined. The EC50 of compound **5** for wild-type SARS-CoV-2 pseudovirus is 14.6 μM for SEMpv and 16.3 μM for Spv, respectively. In contrast, the compound displayed no inhibitory activity against the control VSV-pseudovirus (Gpv). Next, we tested the viral entry activity of compound **5** against different mutant SEMpvs. The data indicated that compound **5** exhibited inhibitory effects on D614G (EC50 12.5 μM) and N501Y (EC50 13.1 μM) (Figure 7A). Furthermore, the activity of compound **5** against virus entry was tested against different variants of the Spv. The data indicated that compound **5** exhibited similar inhibition activity in variants of Alpha (B.1.1.7) (EC50 10.34 μM) and Eplison (B.1.429) (EC50 12.88 μM), displayed better activity in Gamma (P1) (EC50 3.27 μM) and Delta (B.1.617.2) (EC50 3.09 μM), but did not perform as well in Omicron (BA.1) (Figure 7B). These above results demonstrated that compound **5** has the potential to disrupt virus entry across a broad range of SARS-CoV-2 variants. Finally, SARS-CoV-2 (hCoV-19/Taiwan/NTU49/2020) was pre-incubated with compound **5** and the mixture was applied to Calu-3 cells for a pretreatment experiment. The intracellular viral RNA and infectious virus titer were significantly decreased, by 45.6% and 66.3%, respectively, at a concentration of 20 μM (Figure 7C). These data above indicated that compound **5** exhibits anti-viral activity by reducing the entry of SARS-CoV-2.

Compound **5** exhibited higher inhibitory potency against wild-type SARS-CoV-2 than compound **6**, **7**, and **8**, and the results may correlate with their chemical structures. According to the interaction profile generated from iGEMDOCK (Figure 8A), three hydrogen bonds are absent in the docked poses of compound **6**, **7**, and **8**, which interact with the side chain of spike TYR453 and THR500, and the main chain of ASN501. Compounds **5**, **6**, and **8** have the same backbone structure, namely Icosahydro-1H-cyclopenta[a]chrysene (five-ring core, PubChem CID: CID 22085851), but compound **7** does not. The only differences are the functional groups attached to this five-ring core. There are two critical functional groups of compound **5**, but they are absent in compounds **6**, **7**, and **8**. The first is a hydroxymethyl group bound between the cyclopentane and cyclohexane, which interacts with the side chain of THR500 and the main chain of ASN501. The other one is a hydroxyl group bound in one cyclohexane located at the end of this five-ring core, which interacts with the side chain of spike TYR453. These three hydrogen bonds might explain why only compound **5** possesses inhibitory potency against the spike protein in vitro. The 2D interaction plot generated by PoseView [19] is presented in Figure 8B.

To understand the lower inhibition activity of compound **5** in the Omicron variant, multiple sequence alignments of the receptor-binding site of SARS-CoV-2 spike proteins in six variants are displayed. The binding of compound **5** to wild-type strains involves seven positions in the spike proteins, which are marked in blue. However, three residues including G496S, Q498R, and Y505H of Omicron strains occurred in these important binding positions (Figure 9A). It implies that these positions may affect the binding affinity of compound **5** to the spike of Omicron. Figure 9B shows the different docked poses of compound **5** in the wild-type and Omicron strains by the molecular docking method. The receptor-binding domains of wild-type (yellow) and Omicron (pink) spike proteins are superimposed, and the human ACE2 protein structure (PDB ID: 6M0J, chain A), which was complexed with the wild-type spike protein in the crystal structure, is colored in gray. Compound **5**, represented by light green and light purple sticks, was anchored to the spike proteins of both the wild-type and Omicron of SARS-CoV-2, respectively (Figure 9B). The different docked poses of compound **5** in the wild-type and Omicron variants by the molecular docking method may explain why compound **5** did not demonstrate high inhibitory potency against the Omicron strain.

## 3. Discussion

The clinical presentation of COVID-19 varies widely, ranging from asymptomatic cases to fatal outcomes [20]. Though there has been rapid vaccine development, there are still no long-term protective vaccines. Antiviral agents such as protease inhibitors (3CLpro and ritonavir) and polymerase inhibitors (remdesivir and molnupiravir) have shown effectiveness in treating early-stage COVID-19 [21]. However, SARS-CoV-2 elicits an imbalanced host-dependent response in COVID-19 patients. These individuals exhibit elevated levels of proinflammatory cytokines and chemokines, which contribute to pulmonary inflammation and substantial lung damage [22]. Therefore, targeting pro-inflammatory molecules in early phases of COVID-19 is one of the key strategies to preventing the progression of the disease.

There is accumulating evidence demonstrating that the S protein, but not other viral structure proteins of SARS-CoV-2, induces the inflammation response. Stimulation with the S protein induces a polarization of THP-1 macrophages towards a proinflammatory state, characterized by an increase in TNF-α and MHC-II M1-like phenotype markers [23]. In addition, the S protein was thought to be sensed by TLR2, which subsequently activates the NF-κB pathway, resulting in the induction of inflammatory cytokines and chemokines [1]. Our data demonstrated that the S pseudotyped virus induced a similar level of proinflammatory cytokines compared to the SEM pseudotyped virus. Moreover, the inactivated S pseudotyped virus elicited a similar level of inflammatory response as the active pseudotyped virus. These data above support the previous finding that the S protein plays a role in SARS-CoV-2-stimulated inflammation.

CD147 was considered a receptor for SARS-CoV-2, mediating virus entry through endocytosis [4,24]. Our data indicated that the expression level of CD147 on THP-1 cells is higher than ACE2, as observed through fluorescence staining. Moreover, a reduced expression of phosphorylated NFκB and proinflammation cytokine IL-1β was observed in S pesudotyped virus-infected CD147-knockdown THP-1 cells. These data above implied that attachment of spike to receptor or associated factors on the THP-1 cell surface is critical for SARS-CoV-2-induced inflammation. Since there remain some arguments for the roles of CD147 on SARS-CoV-2 infection [4,25,26], the molecular mechanisms underlying spike-induced inflammation in THP-1 cells during SARS-CoV-2 infection deserve further investigation.

It is noted that the chemical structure of compound **5** is similar to betulinic acid, which has been reported to have anti-viral activity by inhibiting SARS-CoV-2 spike binding to ACE2 by computational screening [18]. We therefore tested the inhibition activity of betulinic acid and compound **5** on the entry of SARS-CoV-2. Firstly, we found that compound **5** has significantly higher inhibition activity than betulinic acid. The result is consistent with the structure analysis via iGEMDOCK. Though the two compounds are docked into similar positions of the spike protein (Figure 6A, Table 2), compound **5** has an extra hydrogen bond interaction with residue Y453 and lower binding energy then betulinic acid. It has been reported that Y453, located in the RBD domain of the spike protein, can form hydrogen bonds with H34 in human ACE2 [27]. This suggests that compound **5** possesses a greater potential to disrupt the interaction between the spike and ACE2 due to its higher affinity to the spike protein. In addition, among the similar chemical compounds, compound **5** has also exhibited significant activity against viral entry. Though compound **7** has been identified for its ability to bind to the spike protein [28], compound **5** possesses two essential functional groups that are absent in other compounds. The first group is a hydroxymethyl group located between the cyclopentane and cyclohexane moieties. This group interacts with the side chain of T500 and the main chain of N501. The second one is a hydroxyl group which is attached within a cyclohexane ring located at the terminus of this five-ring core. It engages in interaction with the side chain of spike Y453. The Y453 residue in spike protein RBD is close to His34 in the ACE-2 receptor and T500 and N501 residues in spike protein RBD are close to Tyr41 in ACE-2 [29]. Among these three amino acids, the N501 in wild-type SARS-CoV-2 was found as one of the pivotal residues that increases the binding affinity towards human ACE2 receptors [30]. Therefore, the presence of these three hydrogen bonds could explain the exclusive inhibitory potency of compound **5** against the spike/ACE2-mediated virus entry in vitro.

On the other hand, compound **5** was effective for different variants, except the Omicron variant. Multiple sequence alignment of the receptor-binding site of SARS-CoV-2 spike proteins in six variants showed that seven positions (marked as blue in Figure 8) which participate in compound **5** binding were conserved, excluding the Omicron strain. Three mutations, including G496S, Q498R, and Y505H of Omicron, occurred in these important binding positions. The neighboring residues G496 and F497 of the SARS-CoV-2 spike, as well as residues D355 and Y41 of ACE2, were reported to be crucial for the interaction between the RBD and ACE2 [31]. Our results demonstrated that compound **5** is docked into the spike protein (Figure 6A) and interacts with six residues, which are Y495, G496, Q498, T500, N501, and Y505 (Table 2). However, mutations in G496S, Q498R, and Y505H has been reported to lead to a decrease in the stability of the spike protein [32,33]. It suggests that mutations in these three positions may affect the binding affinity of compound **5** to the spike protein of the Omicron variant and explain why compound **5** did not demonstrate high inhibitory potency against the Omicron strain. These results will provide practical information on developing 3-*epi*-betulin as an adjuvant in current regimens against SARS-CoV-2 in the future.

## 4. Materials and Methods

### 4.1. Purification of Terpenoids from Flueggea virosa and Daphniphyllum glaucescens

Compounds **1**–**3** and **8** were obtained from *F. virosa* [34]. Compounds **1**–**3** were isolated following the methodology outlined in our previous study. Compound **8** (50.0 mg) was obtained from subfraction 5G using an RP-18 column, employing a gradient elution ranging from 75% to 90% MeOH-H_2_O. Compounds **4**–**7** were isolated from *D. glaucescens* [35]. Following the same protocol, we obtained fractions 4G, 4L, and 4N by vacuum liquid chromatography. Fraction 4G was subdivided into 16 subfractions (4G1–4G16) using RP-18 column chromatography (gradient, 30% to 90% MeOH-H_2_O). Subfraction 4G11 was subjected to RP-18 HPLC (90% ACN-H_2_O) to give compounds **4** (2.8 mg) and **5** (217.0 mg). Compound **7** (400 mg) was obtained from fraction 4L utilizing RP-18 column chromatography (95% MeOH-H_2_O). Compound **6** (28.4 mg) was purified from fraction 4N using RP-18 column chromatography with a gradient elution (60–95% MeOH-H_2_O).

### 4.2. Cells, Viruses, and Compounds

THP-1 cells were cultured in RPMI (Gibco, Waltham, MA, USA) supplemented with 10% FBS. BHK-ACE2 cells, a BHK21 cell line stably expressing human ACE2 (gift from Dr. Chia-Yi Yu, National Health Research Institutes, Miaoli, Taiwan), were cultured in RPMI medium supplemented with 5% FBS. Calu-3 cells were grown in Dulbecco’s Modified Eagle Medium (DMEM; Gibco) supplemented with 15% FBS (Gibco), penicillin G sodium 100 units/mL, streptomycin sulfate 100 μg/mL, and amphotericin B 250 ng/mL (antibiotic-antimycotic, Gibco). VeroE6 cells were grown in DMEM supplemented with 10% FBS and 1% antibiotic-antimycotic. All above cells were cultured at 37 °C with 5% CO_2_.

SARS-CoV-2 was isolated from the sputum specimen of COVID-19 patients, and amplified in VeroE6 cells with 2 μg/mL tosylsulfonyl phenylalantyl chloromethyl ketone (TPCK)-trypsin (T1426, Sigma-Aldrich, Darmstadt, Germany). hCoV-19/Taiwan/NTU49/2020 (B.1.1.7, GISAID: EPI_ISL_1010718), hCoV-19/Taiwan/NTU56/2021 (B.1.429, GISAID: EPI_ISL_1020315), hCoV-19/Taiwan/CGMH-CGU-56/2021 (P.1, GISAID: EPI_ISL_2249516), hCoV-19/Taiwan/NTU92/2020 (B.1.617.2, GISAID: EPI_ISL_3979387) and hCoV-19/Taiwan/NTU128/2021 (BA.1, GISAID: EPI_ISL_ 11050301) were the virus isolates used in this study. With the approval of the Institutional Biosafety Committee, all work involving SARS-CoV-2 virus was performed in the Biosafety Level-3 Laboratory of the National Taiwan University College of Medicine. Compounds used in this study are listed in Table 1, with the support of Dr. Chih-Hua Chao (School of Pharmacy, China Medical University, Taichung, Taiwan).

### 4.3. SARS-CoV-2 Pseudovirus

The wild-type SARS-CoV-2 pseudovirus was a lentiviral-based viral particle. The pseudoviral particles of SARS-CoV-2 were generated as described previously [16]. In brief, 293T cells were co-transfected with pcDNA3.1-2019-nCoV-SΔ18 (from RNAi Core, Academia Sinica, Taipei, Taiwan), pCMV d8.91, and pAS3W Fluc, to produce pseudotyped viral particles. The supernatant containing pseudotyped viral particles was collected at 24 h, 36 h, and 48 h post-transfection. The supernatant was then centrifugated at 3000 rpm for 5 min at 4 °C to remove cellular debris following collection by filtration using a low protein-binding filter (0.45 µm) at 4 °C and subsequently ultracentrifugated at 20,000× *g* for 2.5 h. The viral pellets were then resuspended in Opti-MEM medium and stored at −80 °C. The pseudotyped SARS-CoV-2 variants were purchased from RNAi Core (Academia Sinica, Taipei, Taiwan). All SEM pseudoviruses were provided by Dr. Chia-Yi Yu (National Health Research Institutes, Miaoli, Taiwan). The construction and production of the pseudovirus have been described previously [36]. The pseudotyped viral activity was assessed by measuring luciferase activity upon virus entry into cells. The viral particle number was determined by real-time RT-PCR by measuring the RNA copies of FLuc reporter gene.

### 4.4. Real-Time RT-PCR Assay

To determine the gene expression of viral and proinflammation genes, the TRIZOL method was used to extract total RNA from viral transduced cells. The total RNA was reverse-transcribed to cDNA by random primer followed by real-time PCR. In brief, the reaction includes the cDNA, 10 μL of Sybr green master mix (Applied Biosystems), and 250 nM of individual primer, and we adjusted the total volume to 20 μL with reaction conditions as follows: 2 min at 50 °C, 10 min at 95 °C, and then 40 cycles of 15 s at 95 °C and 1 min at 60 °C. To quantify the RNA copies of pseudoviruses, ten-fold serial dilutions of plasmid pAS3W Fluc with known concentration were used to create the standard curve to detect the viral copies of pseudovirus. For SARS-CoV-2 viral RNA, the primer set located in E gene was used to detect the level of viral RNA. The level of glyceraldehyde-3-phosphate dehydrogenase (GAPDH) was used as a control and the relative expression was calculated by the ΔΔCT method [37]. The primers used in this study are listed in Table 3.

### 4.5. Inflammation Induced by SARS-CoV-2 Pseudovirus

The human monocytic cell line THP-1 was treated with 5 ng/mL of phorbol 12-myristate 13-acetate (PMA; Sigma-Aldrich) to differentiate into macrophages in RPMI-1640 medium with 10% FBS for 24 h. After changing fresh medium, the cells were incubated for a further 48 h. The differentiated THP-1 cells were re-seeded into a 12-well plate at a density of 5 × 10^5^ cells per mL, followed by preincubation with or without the tested compound for 2 h prior to SARS-CoV-2 pseudovirus. After 2 h incubation, the cells were washed with PBS and then we changed the fresh medium for a further 16 h incubation. The cell lysates were collected to detect the expression level and phosphorylated status of NF-κB by Western analysis and the RNA was collected to detect the level of viral and proinflammatory genes by RT-qPCR. In parallel, the supernatant was collected to measure the level of IL-1β using Human IL-1β ELISA kit R&D Systems (Minneapolis, MN, USA) according to manufacturer’s instructions. Briefly, 200 μL of the sample was added to each well of an ELISA plate and incubated for 2 h at room temperature (RT). The reaction solution was then aspirated and washed three times. Next, 200 μL of the detection antibody was added to each well and incubated for 1 h at RT. The reaction solution was again aspirated and washed three times. Substrate Solution (200 μL) was added to each well and incubated for 20 min at RT. Finally, Stop Solution (50 μL) was added to each well, and the absorbance was read at 450 nm within 30 min with wavelength correction at 570 nm.

### 4.6. Flow Cytometric Analysis

THP-1 cells were infected with lenti-based shCD147 for 24 h followed by puromycin selection. After 48 h incubation, the expression level of CD147 in the individual group of cells was detected by antibody against CD147 and analyzed by flow cytometry. In brief, the cell pellets from each group of cells were collected by centrifugation at 2000 rpm for 5 min. The cell density was adjusted to 1 × 10^6^ cells/mL in phosphate-buffered saline (PBS) supplemented with 10% goat serum (Gibco). The total volume was maintained at 0.1 mL. The cells were then incubated with the CD147 antibody (Abcam, Cambridge, UK, clone ab666, 1:100) for 30 min at 4 °C. Following that, the cells were incubated with the secondary antibody, Alexa Fluor 488 goat anti-mouse IgG (H + L) (no. A-11029; Invitrogen, Carlsbad, CA, USA), for an additional 30 min in the dark at 4 °C. Flow cytometric analysis was performed using an Attune Nxt FACS (Life Technology/Thermo Fisher Scientific, Waltham, MA, USA).

### 4.7. Immunofluorescence Stain

Calu-3 and differentiated THP-1 cells were fixed with 4% paraformaldehyde for 30 min, followed by three washes with ice-cold PBS. Subsequently, the cells were permeabilized with 0.2% Triton X-100 for 5 min and washed three times with PBS and then blocked by 5% BSA for 1 h. The cells were then incubated with anti-ACE2 antibody (A4612), or anti-CD147 antibody (ab666) followed by staining with Alexa Fluor 488 goat anti-mouse (no. A-11029; Invitrogen, Carlsbad, CA, USA), or AlexaFluor-568 goat anti-rabbit (no. A-11011; Invitrogen, Carlsbad, CA, USA), respectively. Nuclei were stained with DAPI for 10 min, and the cells were washed three times with PBS. Finally, the stained cells were observed under Olympus IX-71 microscope (Olympus Optical, Toyko, Japan).

### 4.8. Cell Viability Assay

Cell viability assay was performed by MTS assay as previously described [38]. In brief, the differentiated THP-1 cells (4 × 10^4^ cells/well), BHK-ACE2 cells (1 × 10^4^ cells/well), and Calu-3 cells (1 × 10^4^ cells/well) were treated with the tested compounds independently at the indicated dose for an additional 48 h. The cells were then incubated with MTS solution for 1 h, and the absorbance at 490 nm was detected by SpectraMax^®^iD3 (Molecular Devices, San Jose, CA, USA).

### 4.9. SARS-CoV-2 Pseudovirus Entry Assay

For pseudoviruses, BHK-ACE2 cells were seeded into 24-well plates (5 × 10^4^ cells/well) and inoculated with 250 μL media containing pseudovirus (5 × 10^6^ particle number). After 2 h incubation, cells were changed with 250 μL fresh media for an additional 48 h inoculation. The cells were then lysed with 50 μL 1 × Passive Lysis Buffer containing 50 μL Bright-glo (Promega, Madison, WI, USA) at room temperature for 15 min. The transduction efficiency was quantified by luciferase activity assay by SpectraMax^®^iD3 (Molecular Devices, San Jose, CA, USA). The IC50 was determined as the concentration at which the response reaches the midpoint of the linear regression.

### 4.10. Plaque Reduction Assay

Plaque reduction assay was performed in 24-well culture plates. The veroE6 cells (ATCC^®^ CRL-1586™) were seeded at 2 × 10^5^ cells/well in DMEM with 10% FBS and antibiotics one day before infection. The cells were then incubated with a different dose of compound for 1 h followed by infection with SARS-CoV-2 (100–150 plaque forming unit (PFU) per well) at 37 °C for 1 h under compound treatment. Subsequently, viruses were removed and the monolayered cells were washed once with PBS before covering with a mixture of 1% methylcellulose and DMEM supplemented with 2% FBS and compound for 120 h. The cells were fixed with 10% formaldehyde overnight. After removal of overlay media, the cells were stained with 0.5% crystal violet and the plaques were counted. The plaque formation ratio was calculated by counting plaque numbers and comparing them to the numbers obtained from the virus control.

### 4.11. Yield Reduction Assay

For pretreatment experiments, virus at MOI with 0.001 was pretreated with a different dose of compound for 1 h at 37 °C. Then, calu-3 cells were infected with mixture for 1 h. After infection, cells were incubated in DMEM supplemented with 2% FBS and 1% antibiotics for 24 h. For post-treatment, calu-3 cells were infected with virus for 1 h, and following that, they were incubated with compound for 24 h. At the end point, culture supernatant was harvested for virus titration, and cell lysate was collected to detect expression of target viral RNA via RT-qPCR.

### 4.12. Virus Titration

The infectious virus titer in the supernatant was determined by plaque assay. Briefly, a total of 2 × 10^5^ veroE6 cells were seeded per well in the 24-well plate one day before being infected for 1 h at 37 °C with SARS-CoV-2. After 1 h, the cells were rinsed with PBS, covered with a mixture of 1% methylcellulose and DMEM supplemented with 2% FBS, and incubated at 37 °C in a 5% CO_2_ incubator. Cells were fixed with 10% formaldehyde for 1 h and stained with 0.5% crystal violet five days later. The plaque formation ratio was then calculated by counting plaque numbers and comparing them to the numbers obtained from the virus or DMSO control. The viral RNA copies in the culture supernatant were quantified by RT-qPCR. Viral RNA in the supernatant was extracted with QIAamp Viral RNA Mini Kit (QIAGEN, Hilden, Germany) according to the manufacturer’s instructions. RNA level of viral E gene in total cellular RNA were determined by RT-qPCR.

### 4.13. Molecular Docking and Interaction Analysis

The molecular docking and interaction analysis between compounds and spike proteins of SARS-CoV-2 were performed by iGEMDOCK [39], a helpful tool for understanding ligand binding mechanisms for a specific protein target. All SARS-CoV-2 spike protein structures of different strains were collected from Protein Data Bank and used as the target proteins. These spike protein structures are listed in Table 4. The figures representing the molecular interaction were all drawn by PyMOL (The PyMOL Molecular Graphics System, Version 2.0, Schrödinger, LLC, New York, NY, USA).

### 4.14. Statistical Analysis

Statistical analysis was assessed by one-way ANOVA (Bonferroni’s Multiple Comparison Test). Data are presented as the mean ± SD of three independent experiments. The *p* < 0.05 was considered statistically significant.

## 5. Conclusions

In summary, we found a compound, 3-*epi*-betulin, isolated from *Daphniphyllum glaucescens*, that exhibits a dual function to inhibit entry of SARS-CoV-2 variants, and spike-induced inflammation by cell models. In addition, 3-*epi*-betulin may play a role as a broad-spectrum compound to inhibit virus entry, diminishing viral load and inflammatory condition in COVID-19. Further investigation is still needed to elucidate the detailed mechanisms by which 3-*epi*-betulin acts against SARS-CoV-2. Through this study, we also define some amino acid residues in the RBD domain that are critical for 3-*epi*-betulin binding. These results may provide new insights for designing compounds subsequently to combat SARS-CoV-2. Overall, our results highlight the potential of 3-*epi*-betulin against SARS-CoV-2.

## Figures and Tables

**Figure 1 ijms-24-17040-f001:**
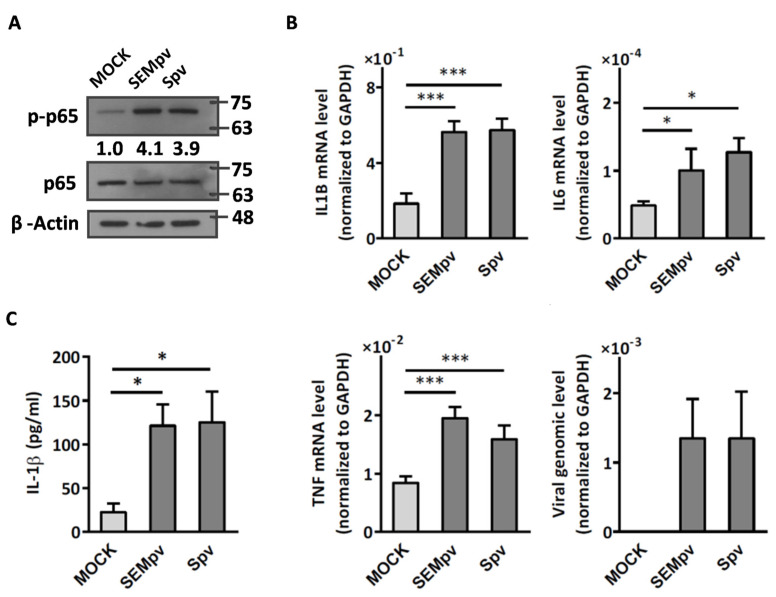
Stimulation of THP-1 cells with SARS-CoV-2 pseudovirus results in an inflammatory response. (**A**) Differentiated THP-1 cells were stimulated by pseudoviruses bearing either SARS-CoV-2 spike, envelope, or membrane protein (SEMpv) for 4 h. Cell lysates were collected and analyzed by Western blotting by indicated antibody. The ratio of phosphorylated p65 versus nonphosphorylated p65 is indicated. β-Actin was used as a normalized control. (**B**) THP-1 cells were stimulated for 16 h using pseudoviruses, and the expression levels of pro-inflammatory genes (IL1B, IL6, and TNF) and viral RNA were detected by RT-qPCR. GAPDH was used as a normalized control. (**C**) In parallel, the concentration of IL-1β cytokine was detected by ELISA in the supernatant. Data are presented as the mean ± S.D. (*n* = 3; *, *p* < 0.05; ***, *p* < 0.001).

**Figure 2 ijms-24-17040-f002:**
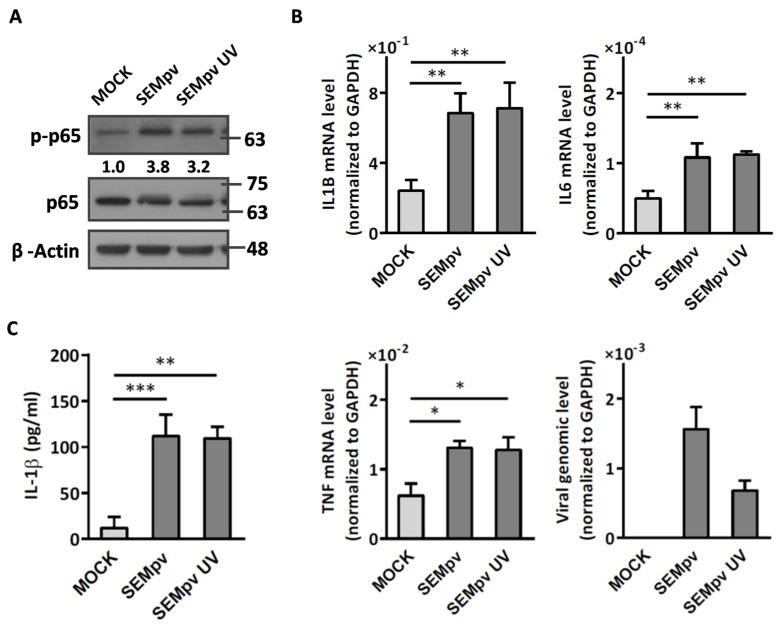
The structural proteins of SARS-CoV-2 pseudovirus induce an inflammatory response. (**A**) Differentiated THP-1 cells were stimulated with either UV-inactivated or active SARS-CoV-2 pseudovirus for 4 h, and the cell lysates were collected for Western blotting by indicated antibodies. (**B**) Differentiated THP-1 cells were stimulated with either UV-inactivated or active SARS-CoV-2 pseudovirus for 16 h, and the cellular RNA was collected to detect the expression of pro-inflammatory genes (IL1B, IL6, and TNF) and viral RNA by RT-qPCR. GAPDH was used as a normalized control. (**C**) The concentration of IL-1β cytokine in the supernatant was analyzed by ELISA. Data are presented as the mean ± S.D. (*n* = 3; *, *p* < 0.05; **, *p* < 0.01; ***, *p* < 0.001).

**Figure 3 ijms-24-17040-f003:**
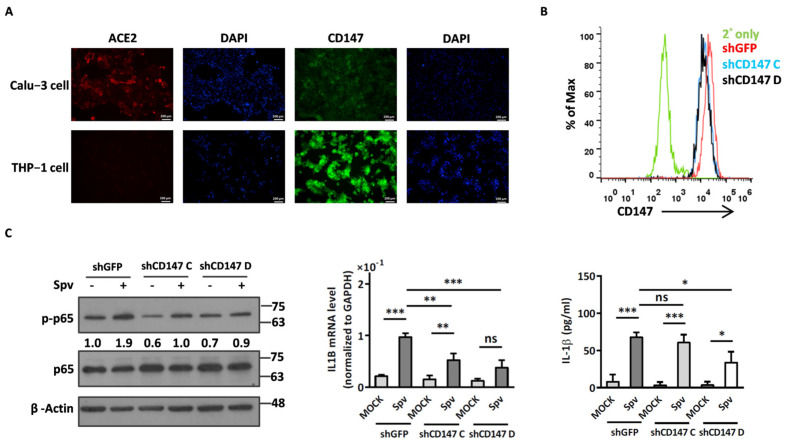
CD147 facilitates SARS-CoV-2-induced inflammation response in differentiated THP-1 cells. (**A**) Expression of ACE2 and CD147 in Calu-3 and differentiated THP-1 were detected by immunofluorescence staining with indicated antibody. DAPI stain as a nuclear marker. (**B**) THP-1 cells were knocked down by lenti-based shCD147 for 48 h and the CD147 expression level was evaluated using antibody detection and analyzed by flow cytometry. (**C**) CD147-knockdown THP-1 cells were stimulated by SARS-CoV-2 pseudotyped virus. The protein lysates were collected after 4 h virus incubation and the Western blotting was used to detect the phosphorylation status of p65 by indicated antibody (**left panel**). The RNA was collected after 16 h stimulation, and the level of proinflammation gene IL1b was detected by RT-qPCR (**middle panel**) and the level of IL1B cytokine in supernatant was detected by ELISA (**right panel**). Data are presented as the mean ± S.D. (*n* = 3; *, *p* < 0.05; **, *p* < 0.01; ***, *p* < 0.001; ns, not significant).

**Figure 4 ijms-24-17040-f004:**
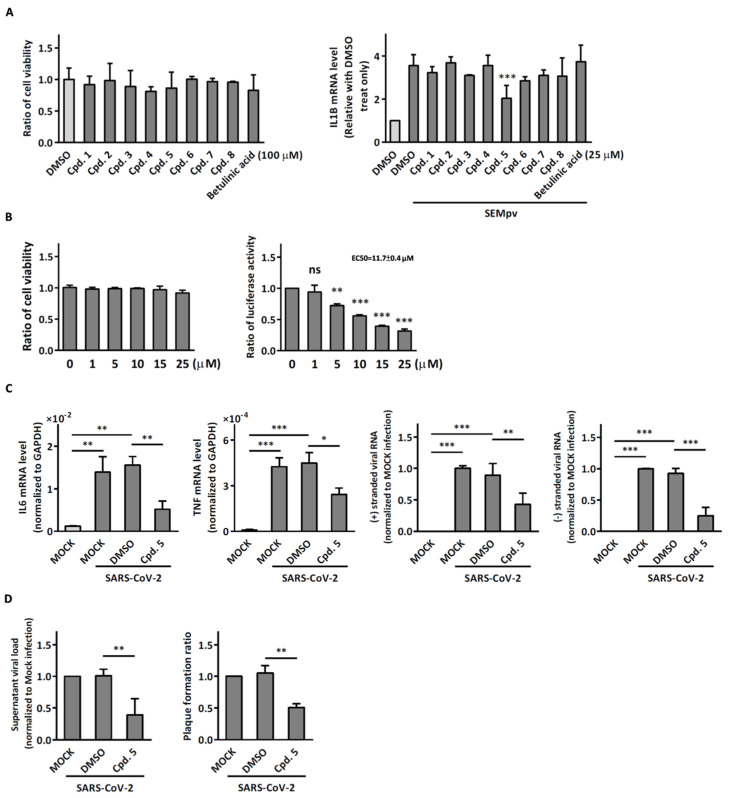
Compound **5** inhibits SARS-CoV-2-induced inflammation and viral replication. (**A**) Differentiated THP-1 cells were treated with potential anti-inflammation compounds for 48 h. The cell viability was determined by MTS assay (**left panel**). Differentiated THP-1 cells were infected with SARS-CoV-2 pseudovirus and treated with potential anti-inflammation compounds for 16 h. Gene expression of IL-1B was detected by RT-qPCR (**right panel**). (**B**) 293T cells were treated with indicated dose of compound **5**. After 48 h, the cell viability was determined by MTS assay (**left panel**). 293T cells were transfected with pBAC-SARS-CoV-2 bearing luciferase gene and pCAG2-NP-HA. After 5 h, the cells were reseeded and compound **5** was added into the culture medium for 24 h. Viral replication was detected by luciferase activity assay, while cell viability was determined using an MTS assay. The results are presented as the ratio of luciferase activity to cell viability (**right panel**). (**C**) Calu-3 cells were infected with SARS-CoV-2 (hCoV-19/Taiwan/NTU49/2020) at MOI of 0.005 for 1 h. After infection, the cells were treated with either DMSO or 10 μM compound **5** for 24 h. The cell lysate was collected to detect the expression levels of pro-inflammatory genes (IL-6 and TNF) and different sense of viral RNA via RT-qPCR. (**D**) The culture supernatant was then harvested to determine the virus titer using RT-qPCR and plaque assay. Data are presented as the mean ± S.D. (*n* = 3; *, *p* < 0.05; **, *p* < 0.01; ***, *p* < 0.001; ns, not significant).

**Figure 5 ijms-24-17040-f005:**
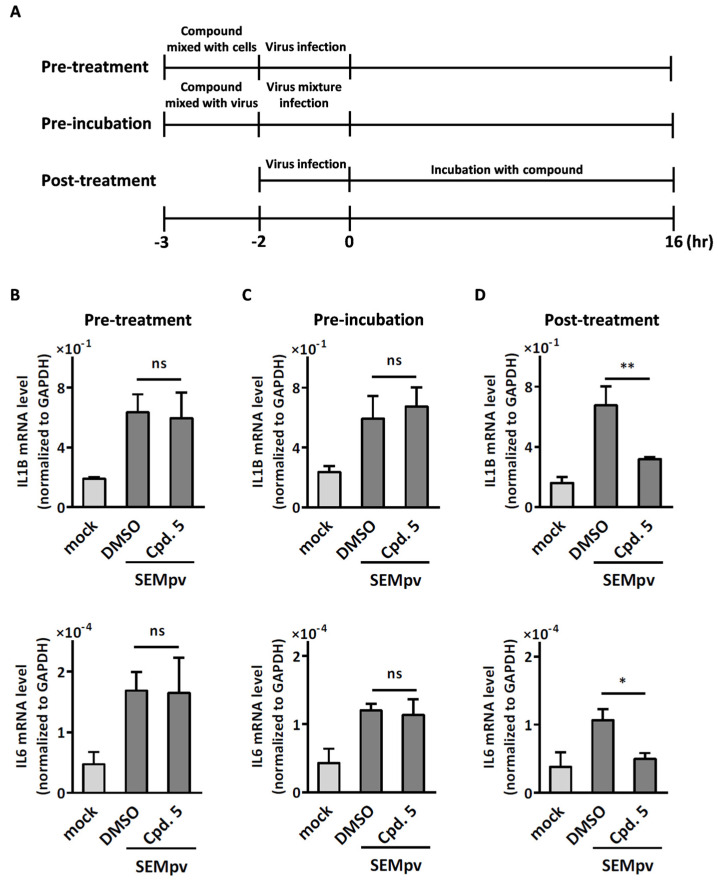
Compound **5** inhibits SARS-CoV-2-induced inflammation not related to virus entry. (**A**) Schematic representation of the timeline to evaluate the anti-inflammatory activity of compound **5**. After 16 h incubation, cellular RNA was collected, and the expression levels of pro-inflammatory genes were determined by RT-qPCR. (**B**) Differentiated THP-1 cells were treated with compound **5** for 1 h before infection with SARS-CoV-2 pseudovirus. (**C**) Compound **5** was mixed with SARS-CoV-2 pseudovirus for 1 h prior to infecting the differentiated THP-1 cells. (**D**) Differentiated THP-1 cells were infected with SARS-CoV-2 pseudovirus and subsequently treated with compound for 16 h. GAPDH was used as a normalization control. The data are presented as the expression ratio of the inflammatory gene to GAPDH. Data are presented as the mean ± S.D. (*n* = 3; *, *p* < 0.05; **, *p* < 0.01; ns, not significant).

**Figure 6 ijms-24-17040-f006:**
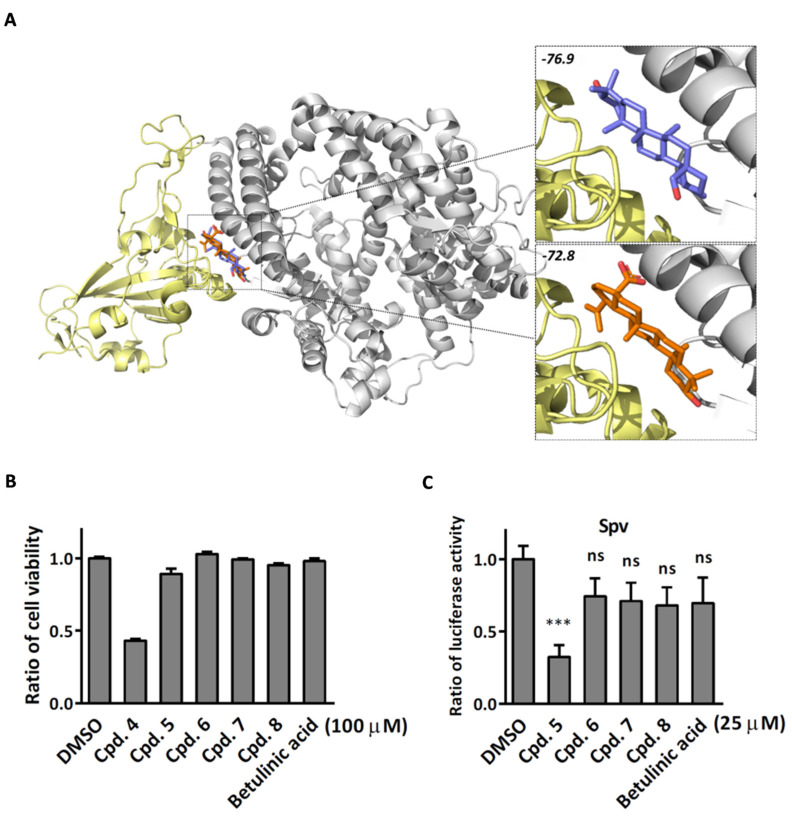
Compound **5** has the potential to inhibit virus entry. (**A**) The docked poses of compound **5** and *betulinic acid* anchored with the wild-type SARS-CoV-2 spike proteins. The receptor-binding domain of wild-type spike proteins (yellow) was complexed with the human ACE2 protein structure (PDB ID: 6M0J, chain A, gray). Two compounds, compound **5** (light purple) and betulinic acid (orange), docked with wild-type spike proteins by iGEMDOCK, have −76.9 and −72.8 kcal/mol binding energy, respectively. (**B**) BHK-ACE2 cells were treated with indicated compounds for 48 h. Cell viability was then determined by MTS assay. (**C**) SARS-CoV-2 pseudoviruses were pretreated with DMSO or each of indicated compound at 37 °C for 1 h. Subsequently, BHK-ACE2 cells were incubated with the compound–virus mixture for additional 48 h followed by luciferase activity assay. (*n* = 3; ***, *p* < 0.001, ns, not significant).

**Figure 7 ijms-24-17040-f007:**
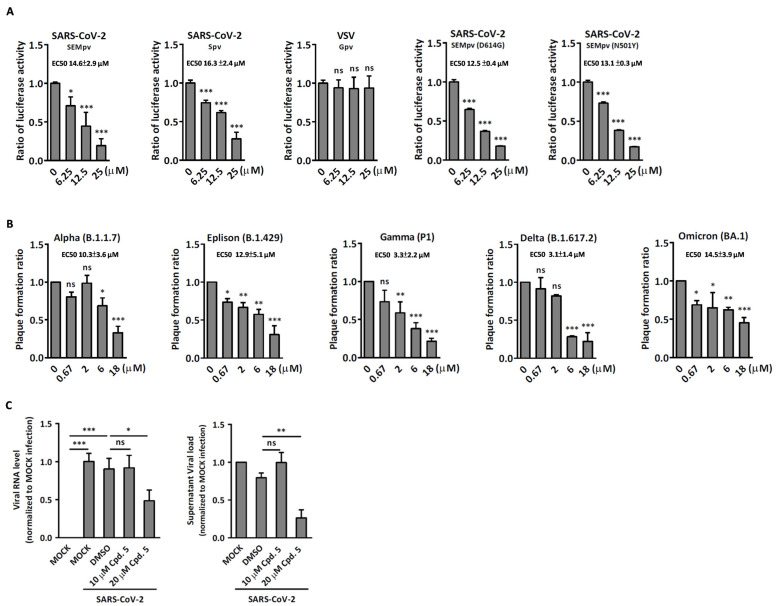
Compound **5** exhibits inhibitory effects on virus entry. (**A**) Pseudovirus bearing spike from different variants of SARS-CoV-2 was pre-incubated with serially diluted compound at 37 °C for 1 h. The mixture was then put into BHK-ACE2 cells for 48 h followed by luciferase activity assay to determine the EC50 of the compound to each variant of pseudovirus. (**B**) Each of SARS-CoV-2 variants was pre-incubated with a serial dilution of the compounds at 37 °C for 1 h. Subsequently, VeroE6 cells were incubated with the compound–virus mixture for an additional hour, followed by incubation without compound treatment for 120 h. The plaque formation ratio was calculated by counting the number of plaques and normalizing them to the numbers obtained from the DMSO control. (**C**) SARS-CoV-2 (NTU49) with MOI at 0.001 was pretreated with DMSO, 10 μM, or 20 μM compound at 37 °C for 1 h. Calu-3 cells were then incubated with the compound–virus mixture for an additional 1 h, followed by incubation without compound treatment for 24 h. The culture supernatant was collected, and the virus titer was determined using qRT-PCR (**left**) and plaque assay (**right**). Data are presented as the mean ± S.D. (*n* = 3; *, *p* < 0.05; **, *p* < 0.01; ***, *p* < 0.001; ns, not significant).

**Figure 8 ijms-24-17040-f008:**
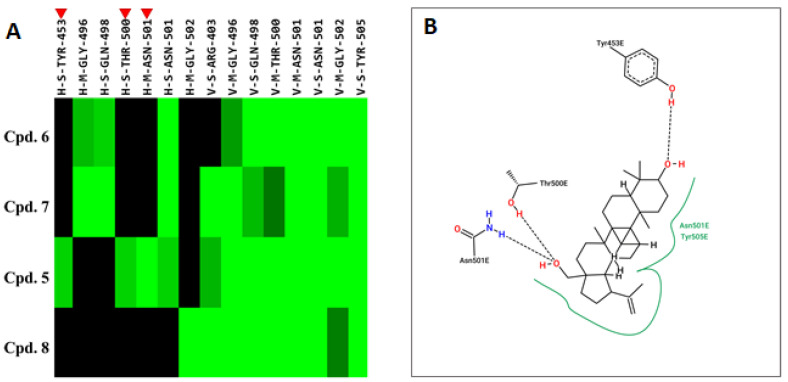
Interaction profile and 2-D interaction plot of compound **5** binding to wild-type SARS-CoV-2 spike protein. (**A**) The interaction profiles of compound **5**, **6**, **7,** and **8** by iGEMDOCK. Each interactive residue listed on the x-axis has four codes separated by hyphens. The first code of the interactive residue stands for the force between compounds and residues, H for hydrogen bond force, and V for van der Waals forces. The second code represents the interaction in the main chain (M) or side chain (S). The third code represents the spike protein’s residue type, and the fourth code is the residue serial number. The interactions are represented in green when the energy ≤ –2.5 for H and the energy < –4 for V. The absent interactions are colored in black. Three red triangles indicate these interactive residues interact with compound 5 but not with other compounds. (**B**) 2-D interaction plot of the interactions between docked compound **5** and the wild-type spike protein visualized by PoseView.

**Figure 9 ijms-24-17040-f009:**
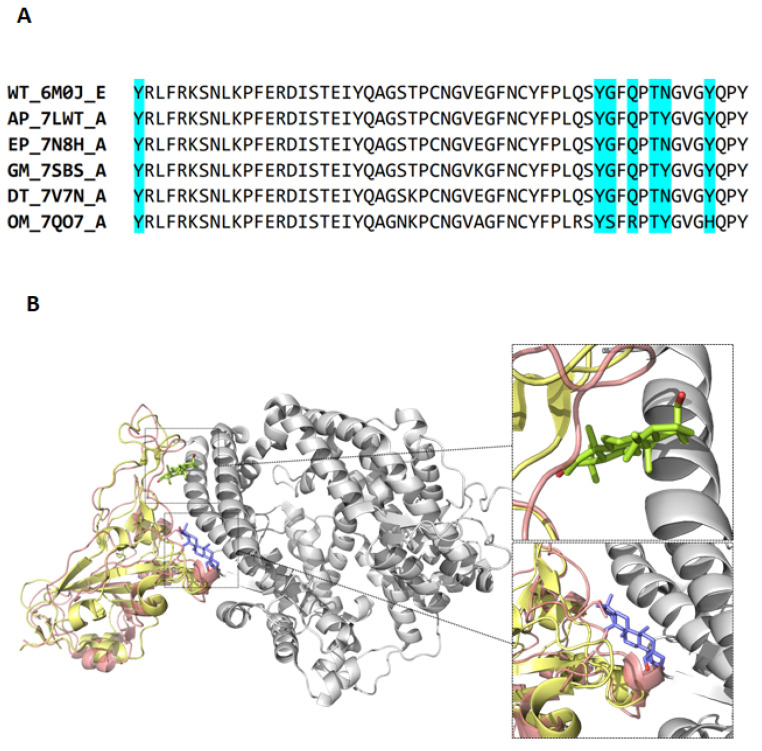
Sequence alignment and structural comparison of compound **5** binding to SARS-CoV-2 spike protein. (**A**) The multiple sequence alignment of the receptor-binding domain in the SARS-CoV-2 spike proteins was presented. The title of each protein sequence is in the format “ Abbreviations_PDB ID_Chain” listed in table in Section 4.13. The amino acid residues that compound bind to spike protein are marked blue. (**B**) The receptor-binding domains of wild-type (yellow) and Omicron (pink) spike proteins are superimposed. The human ACE2 protein structure (PDB ID: 6M0J, chain A), which was complexed with the wild-type spike protein in the crystal structure, is colored in gray. Compound **5,** with light green or light purple sticks, is indicated as anchored with wild-type or Omicron strains of SARS-CoV-2 spike proteins, respectively.

**Table 1 ijms-24-17040-t001:** Compounds used in this study.

No.	Compound Name	Compound Structure
Cpd. **1**	**12-Hydroxy-13-methyl-*ent*-podocarp-6,8,11,13-tetraen-3-one**	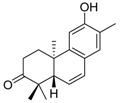
Cpd. **2**	**3β,12-Dihydroxy-13-methylpodocarpa-6,8,11,13-tetraene**	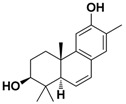
Cpd. **3**	***ent*-3β,12α-Dihydroxypimara-8(14),15-diene**	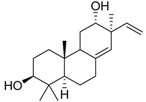
Cpd. **4**	**3-*epi*-Betulinic acid**	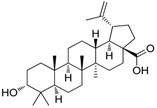
Cpd. **5**	**3-*epi*-Betulin**	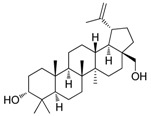
Cpd. **6**	**Lup-20(30)-ene-3β,29-diol**	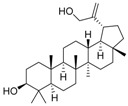
Cpd. **7**	**Betulin**	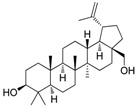
Cpd. **8**	**Lup-20(29)-ene-1,3-dione**	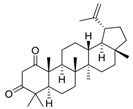

**Table 2 ijms-24-17040-t002:** Amino acid residues of wild-type SARS-CoV-2 spike protein interact with 3-*epi*-betulin and betulinic acid.

Compound	Energy	Binding Residues
Cpd. **5**	−76.9	Y453, Y495, G496, Q498, T500, N501, Y505
Betulinic acid	−72.8	Y495, G496, Q498, T500, N501, Y505

**Table 3 ijms-24-17040-t003:** Primers used in this study.

Target	Forward Primer (5′-3′)	Reverse Primer (5′-3′)
GAPDH	TGGGTGTGAACCATGAGAAG	GCTAAGCAGTTGGTGGTGC
FFLuc	ATTACACCCGAGGGGGATGA	CCAGATCCACAACCTTCGCT
IL1B	GGACAAGCTGAGGAAGATGC	GATTCTTTTCCTTGAGGCCC
IL6	AATGAGGAGACTTGCCTGGT	GCAGGAACTGGATCAGGACT
TNF	CTGCACTTTGGAGTGATCGG	AGGGTTTGCTACAACATGGG
RT primer for (+) stranded viral RNA	AGAAGGTTTTACAAGACTCACGTT	
RT primer for (−) stranded viral RNA	CGAACTTATGTACTCATTCGTTTCGG	
SARS-CoV-2_E	ACAGGTACGTTAATAGTTAATAGCGT	ATATTGCAGCAGTACGCACACA

**Table 4 ijms-24-17040-t004:** The SARS-CoV-2 spike protein structures of different strains used in molecular docking and interaction analysis.

Strain	Abbreviations	PDB ID	Chain	Reference
Wild-type	WT	6M0J	E	[40]
Alpha (B.1.1.7)	AP	7LWT	A	[41]
Epsilon (B.1.429)	EP	7N8H	A	[42]
Gamma (P1)	GM	7SBS	A	[43]
Delta (B.1.617.2)	DT	7V7N	A	
Omicron (BA.1)	OM	7QO7	A	

## Data Availability

All relevant data are included in the paper.

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
