# Peer review of "Dual Effects of 3-epi-betulin from Daphniphyllum glaucescens in Suppressing SARS-CoV-2-Induced Inflammation and Inhibiting Virus Entry"

_ijms, 2023, doi:10.3390/ijms242317040_

Round 1

Reviewer 1 Report

Comments and Suggestions for Authors

The manuscript by Yeh et al. describes how 3-epi-betulin has anti-inflammatory properties relevant to SARS-CoV-2, and ability to block SARS-CoV-2 viral replication in host cells.  The activities are apparently due to 3-epi-betulin binding the viral spike protein that facilitates interaction with host cells.  Evidence for the enhanced efficacy of 3-epi-betulin versus structurally related compounds is predicted via molecular docking utilizing available crystal structures.  I find the manuscript well organized and the experimental designs well suited to support the objectives.  The data also support continued investigation of 3-epi-betulin as an anti-SARS-CoV-2 agent.  However, discrepancies both within the paper and the literature need to be reconciled (see Major Comments), and clarity should be enhanced in the Results section (see Minor Comments).

Major comments:

Anti-inflammatory and anti-viral properties of 3-epi-betulin (also called Compound 5) are established in THP-1 cells (Figure 4A) and 293-T cells (Figure 4B) respectively. However, these results may be explained by toxicity (either cell type) or proliferation (293-T).  Please show effects of 3-epi-betulin on cell viability and proliferation in THP-1 and 293-T.  Indeed the authors show compound effects on viability in BHK-ACE2 cells in Figure 6 for anti-viral, but not in cell types used in Figure 4 for anti-inflammatory. 

The authors use reference 16 to indicate betulinic acid inhibits SARS-CoV-2 by binding to viral spike protein. However, they demonstrate that betulinic acid does not inhibit viral replication in their Luciferase based model that relies on viral spike protein in Figure 6C.  This contradicts their predicted binding of betulinic acid to viral spike protein (Table 2).  Which is correct: reference 16 and Table 2 (does inhibit viral entry & replication) or the authors’ data in Figure 6C (does not inhibit viral replication)?

Similarly, betulin (compound 7) was predicted to bind SARS-CoV-2 spike protein (https://link.springer.com/article/10.1007/s11224-022-02079-8).  Please address the discrepancy between this compound binding to spike protein, but no efficacy in the replication assay in Figure 6C.

Minor comments.

Optional: Please add Compound 8 and betulinic acid to the THP-1 cell screening assay in Figure 4.  Are the anti-inflammatory properties of 3-epi-betulin unique amongst the structurally related compounds in this study?

The last 2 sentences of the Introduction should specify that anti-SARS-CoV-2 properties of 3-epi-betulin were observed in cell culture.  As written, the statements suggest in vivo or clinical findings. 

Please double check: only 1 “*” for Figure 1C left panel?

The source of the compounds used in this study needs to be added.  Referencing the authors’ “compound library” is not adequate.

For clarity, please add a statement in section 2.2 to introduce the replication assay.

What is the concentration of the compounds used in Figure 4A?

In Figure 4 C & D “mock” may be misused.  Mock usually means no virus.  Do you mean no treatment, such as no DMSO or any other compound?

Data from Figure 5 are derived from pre-treating THP-1 cells with compound 5.  Data from Figure 6C and Figure 7 are derived from pre-treating the virus with compounds, presumably to allow compound to bind spike protein before treating human cells.  This is confusing, so please add statements in the Results section specifying what is being pre-treated.

Add a statement that IC50s determined in Figure 7 are from a limited 4 to 5 point curves.  Also, the number of decimal places in Figure 7A and B are not justified and lack units (micromolar).

Add how IC50s were determined. 

The title of Table 2 states Botulin, which is the corresponding alcohol of Betulinic acid shown in the Table.   Please correct the title of Table 2.

Please reference Table 4 in the legend of Figure 9A. It is currently not clear what the text on the left of Figure 9A refers to.

Evidence for betulin (compound 7) and betulunic acid to inhibit the SARS-CoV-2 or SARS-CoV protease 3CLpro has been published (https://www.mdpi.com/1420-3049/26/9/2654#B12-molecules-26-02654 https://link.springer.com/article/10.1007/s11224-022-02079-8 https://pubs.acs.org/doi/10.1021/jm070295s).  The authors are encouraged to Discuss the potential for the structurally related 3-epi-betulin to have a triple mechanism of action for anti-SARS-CoV-2 based on these publications.

SARS-CoV-2 is frequently misspelled, such SARS-CoV2 found 8 times or SARS CoV2 in the last sentence of Introduction.

Errors in gene name versus protein name are common, particularly with IL1B and TNF (not TNFA).  Please correct these errors throughout the manuscript.

Comments on the Quality of English Language

Overall, the manuscript is well written and is clear.  However, errors in grammar are present throughout this manuscript, such as the title of Figure 2, lines 114 to 115, line 317 than versus then, etc. Please have the manuscript edited for grammar.

Reviewer 2 Report

Comments and Suggestions for Authors

The manuscript entitled "Dual effects of 3-epi-betulin from Daphniphyllum glaucescens in suppressing SARS-CoV-2-induced inflammation and inhibiting virus entry" is a good piece of work. However, several questions were raised. How does the authors find that epi-betulin showed that activity? randomly what was the rational? because it is a natural product? please expand. How the authors choose the compounds tested? just because they have them? reasons? Were all compounds listed isolated from the plant? There is no source of the compounds? what about the purity of the compounds? If the compounds were isolated from the plant where is the experimental of these? 

As the authors concluded, If the compound 5 or epi-betulin is not active in the latest omicron strain what is the benefit of this research? please discussed. 

The manuscript needs to add several other details to make it reproducible. 

Comments on the Quality of English Language

Several grammatical errors, please recheck.

Round 2

Reviewer 2 Report

Comments and Suggestions for Authors

The authors had been addressed all the comments and suggestions. The manuscript is now acceptable.
